

**European sulphate aerosols were a key driver of the early**
**twentieth-century intensification of the Asian summer monsoon**
Weihao Sun[1,2], Massimo A. Bollasina[1], Ioana Colfescu[3,4], Guoxiong Wu[2] and Yimin Liu[2]
[1]School of GeoSciences, University of Edinburgh, Edinburgh, UK
[2]Key Laboratory of Earth System Numerical Modelling and Application, Institute of Atmospheric Physics,
Chinese Academy of Sciences, Beijing, China.
[3]School of Mathematics and Statistics, University of St Andrews, St Andrews, UK
[4]National Centre for Atmospheric Science, UK
Correspondence to: Massimo A. Bollasina (Massimo.Bollasina@ed.ac.uk)
**Abstract.** Observations show that the Asian summer monsoon experienced substantial multi-decadal changes
during the early 20th century, including a wetting trend over South Asia and a southward rainfall shift over East
Asia. Despite their significance, these variations have received limited attention, and the underlying mechanisms
remain poorly understood. This study investigates the role of increased European sulphate aerosol emissions in
shaping these monsoon changes using ensemble experiments with the Community Earth System Model. The
aerosol-driven rainfall patterns over South and East Asia resemble observations, suggesting that European
aerosols played an important role in modulating the monsoon. These changes are linked to large-scale anomalies
in surface climate and three-dimensional atmospheric circulation across the Indo-Pacific, which alter moisture
transport to the continent, the main driver of the rainfall anomalies. Regional circulation anomalies form part of a
hemispheric upper-tropospheric wave train originating over central Europe and extending through the Middle East
to the Pacific. The wave train arises as a thermodynamic adjustment to the aerosol-induced surface cooling and
related anticyclone over Europe, extends to the upper troposphere, and, while propagating eastward, induces three-
dimensional circulation anomalies across Asia that affect the monsoon. These findings provide compelling
evidence for the influence of European sulphate aerosols on the early 20th-century monsoon variability, which is
relevant for improving current understanding of the regional-scale impacts of anthropogenic aerosols. As
European $SO_2$ emissions continue to decline, this study sheds light upon a possible ongoing and future pathway
which may significantly modulate the monsoon response to Asian aerosol changes.





## 1 Introduction

The Asian summer monsoon (ASM) is a vital source of water for over 60% of the world's population (Turner and Annamalai, 2012). Its interannual variability and long-term changes have significant implications for water resources, agriculture, and economic activities across Asia (e.g., Gadgil and Rupa Kumar, 2006).

In the past decade, anthropogenic aerosols have attracted considerable scientific interest due to their ability to offset part of the warming caused by greenhouse gases (GHGs) (e.g., Samset et al., 2018; Hegerl et al., 2019; Li et al., 2022). Aerosols influence climate by scattering and absorbing solar radiation, and by acting as cloud condensation and ice nuclei, thereby modifying cloud albedo, lifetime, and precipitation processes (Boucher et al., 2013; Stier et al., 2024). They remain the largest source of uncertainty in estimating anthropogenic climate forcing since the pre-industrial era (Andrews and Forster, 2020; Bellouin et al., 2020). While their global historical effective radiative forcing is smaller than that of GHGs, aerosols can trigger strong regional climate responses due to their spatial and temporal variability (e.g., Szopa et al., 2021).

The link between long-term changes in aerosol emissions and ASM variability is widely debated (e.g., Lau 2016; Li et al., 2016; Wu et al., 2016). Research is particularly challenging due to the compounding effects of internal variability, model biases, and divergent model responses (e.g., Saha and Gosh, 2019; Liu et al., 2024). Nonetheless, studies using observational and modelling evidence have highlighted the significant impact of aerosols, both regional (i.e., Asian-only) and remote (i.e., outside Asia), on the late 20th century weakening of the South ASM and the emergence of a southern-flood-northern-drought (SFND) pattern over East Asia (e.g., Bollasina et al., 2011; Guo et al., 2013; Salzmann et al., 2014; Li et al., 2015; Dong et al. 2019; Wilcox et al. 2020).

Gauge records reveal clear multi-decadal variability in both South Asian and East Asian summer monsoons throughout the 20th century (e.g., Zhang and Zhou, 2011; Preethi et al., 2017; Li et al., 2023). In particular, the first half of the century saw an increase in rainfall over South Asia, most notably in central India, and a tripole pattern over East Asia, with excess rainfall in the south and northeast, and drier conditions along the Yangtze River basin. These anomalies appear to be part of a coherent ASM-wide fluctuation, mirrored in broader NH summer monsoon changes (Zhang and Zhou, 2011; Preethi et al., 2017; Wang et al., 2018; Goswami et al., 2023).


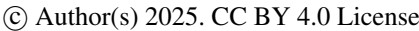

Strikingly, this pattern reversed in the second half of the 20[th] century, producing comparable but opposite sign
anomalies, and resulting in minimal net change over the full period.

While monsoon changes since the 1950s have been extensively studied (e.g., Bollasina et al., 2011; Salzmann et
al., 2014; Song et al., 2014; Liu et al., 2019; Liu et al., 2024; Shao et al., 2024), early-century variations have
received less attention. Yet, given the contrasting trends throughout the historical period, understanding monsoon
drivers and the underpinning physical mechanisms in the entire record is essential to constrain the nature of its
multi-decadal variability and improve future projections. This is particularly relevant in the context of
disentangling internal climate variability from externally-forced changes (e.g., Salzmann and Cherian, 2015;
Huang et al., 2020). In this regard, compared to the later period, it is conceivable to assume a negligible role of
Asian aerosols in the first half of the 20[th] century, as emissions underwent a significant growth only from the
1950s and a sharp rise after the 1970s (Smith et al., 2011; Lund et al., 2019). In contrast, North American and
European aerosols had already increased substantially, raising the possibility of their significant influence on the
monsoon.

Several studies have highlighted the potential role of remote, including European, aerosols in driving monsoon
changes (e.g., Cowan and Cai, 2011; Bollasina et al., 2014; Guo et al., 2015; Dong et al., 2016; Liu et al., 2018;
Shawki et al., 2018; Undorf et al., 2018a,b; Westervelt et al., 2018; Wang et al., 2020). However, these analyses
often relied on idealised simulations, such as long equilibrium experiments with present-day emissions turned on
or off. Whether aerosol influences were already detectable in the early 20[th] century remains an open question.
Addressing this gap is crucial for better understanding and narrowing uncertainty in aerosol-related future
monsoon projections, especially considering divergent present-day emission trends (e.g., Wang et al., 2021a) and
a wide range of plausible future pathways (Samset et al., 2019; Persad et al., 2023).

Against this backdrop, this study investigates whether rising European sulphate aerosols – the dominant
anthropogenic aerosol species of the region (e.g., Hoesly et al., 2018) – had any detectable impact on the ASM
during the first half of the 20[th] century. Section 2 introduces the climate model, experimental setup, and
observational data used. Section 3 presents the ASM response and examines the underlying mechanism.
Discussion and conclusions are provided in Section 4.




**2 Data and methods**
The primary data comprises output from two transient historical (1850–2005) experiments with the fully coupled
Community Earth System Model (CESM) version 1.2.2 (Hurrell et al., 2013). The atmospheric component is the
Community Atmospheric Model version 5.3 (CAM5.3, Neale et al., 2012), which includes a 3-mode aerosol
scheme and a prognostic representation of aerosol-cloud interactions (Ghan et al., 2012). The horizontal
resolutions are 1.9° x 2.5° for the atmosphere and 0.6° x 0.9° for the ocean. The model reproduces spatial patterns
and magnitude of the climatological summertime monsoon rainfall and low-tropospheric circulation reasonably
well (Supplementary Fig. S1), consistent with the overall performance of CMIP5/CMIP6 models (Meehl et al.,
2020; He et al., 2023). A full description of the model and experiments is provided by Undorf et al. (2018a); here,
we briefly summarise key details.

Each experiment consists of an 8-member ensemble initialised from different points in a 200-year pre-industrial
(PI) simulation with fixed anthropogenic emissions at 1850 levels. The first experiment (ALL) includes time-
varying historical emissions of anthropogenic aerosols, greenhouse gases, and natural forcing factors. The second
(fixEU) is identical to ALL except that anthropogenic $SO_2$ and $SO_4$ emissions over Europe (EU) are held at PI
levels. Europe is defined according to Tier 1 regions from the Hemispheric Transport of Air Pollution 2
experiments (Koffi et al., 2016).

Several observational datasets are also used. Precipitation was taken from three land-only monthly gridded gauge-
based datasets to account for uncertainties and discrepancies: the 1° x 1° Global Precipitation Climatology Centre
v2020 (GPCC; Schneider et al., 2014), the 0.5° x 0.5° Climate Research Unit v4.00 (CRU; Harris et al., 2014),
and the 0.5° x 0.5° University of Delaware v401 (UDEL, Willmott and Matsuura, 1995). Monthly sea level
pressure surface is from the 5° x 5° Hadley Centre Sea Level Pressure dataset (HadSLP2; Allan and Ansell, 2006),
and 850-hPa winds from the ECMWF Reanalysis 5 (Hersbach et al., 2020). Monthly observed precipitation from
the 2.5° x 2.5° Climate Prediction Center Merged Analysis of Precipitation dataset (CMAP, Xie and Arkin, 2017)
is also used for model validation.



The analysis focuses on the summer (June–August, JJA) monsoon variation during the early 20th century.
Temporal changes are estimated using least-squares linear trends from 1901 to 1955. Alternative trend estimation
methods (i.e., Sen's non-parametric slope or differences between the 1941–1955 and 1901–1915 means) yield
similar results (not shown). As the main objective is to isolate the influence of external forcing, particularly
anthropogenic aerosols, on the ASM, changes are primarily assessed using ensemble means. Assuming linear
additivity of responses, the difference between the ALL and fixEU experiments is interpreted as the impact of
European $SO_2$ emissions. Statistical significance is assessed using a two-tailed Student's $t$-test at the 90%
confidence level. To identify the contribution of the model's internal variability to the forced response, individual
ensemble members are also examined. Additionally, the PI simulation is used to assess the statistical significance
of the forced changes relative to internal variability. We generate 10,000 bootstrap samples of 8-member ensemble
means to construct the probability distribution of unforced linear trends; the 90% confidence interval is defined
as the range within which 90% of these fall.
**3 Results**
**3.1 Observed and simulated precipitation trends**
Observational data show that JJA precipitation changes across Asia during the first half of the twentieth century
exhibit a coherent large-scale pattern (Fig. 1a), with robust features across multiple datasets (Supplementary Fig.
S2). Over South Asia, significant wetting is observed across central and northern India, contrasted by drying over
southern and northeastern India, the eastern Himalayas and northern Myanmar. Further east, rainfall deficits occur
over southern Indochina and Indonesia, with a pronounced drying over the middle and lower reaches of the
Yangtze River valley (105°–120°E, 25°–35°N). In contrast, precipitation increases are seen in a band extending
from northern Indochina through southern China, and further north into northwestern China and the Korean
peninsula. Interestingly, the anomalous rainfall pattern over South Asia closely resembles, but with opposite sign,
that associated with the late 20th century weakening of the monsoon, while the anomalies over eastern Asia are
reminiscent of the reversed SFND pattern (e.g., Bollasina et al., 2011; Dong et al., 2016).

To provide further context on the long-term precipitation variations, Figure 1c shows the observed time series of
monsoon rainfall anomalies over the core Indian monsoon region (75°–87°E, 16°–27°N land-only points, see box
in Fig. 1c). Beyond interannual and decadal fluctuations, the record reveals a marked increase from the 1900s to





147 the mid–1950s (+0.55/+0.76 mm day$^{-1}$ in GPCC/CRU over 55 years, statistically significant at the 90% confidence

148 level), equivalent to about 7–11% of the long-term climatology. The anomaly also results in a 5–7% increase in

149 the All–India rainfall relative to early–century values. A similar analysis over southern China (110°–120°E, 25°–

150 35°N; Supplementary Fig. S3) also shows a prominent, statistically significant, multi-decadal trend.

151

152 Despite its coarser resolution, the ALL ensemble mean captures the main spatio-temporal characteristics of the

153 observed precipitation trends (Fig. 1b). The model reproduces the widespread precipitation increase over northern

154 India, albeit with a weaker magnitude (+0.14 mm day$^{-1}$ (55 years)$^{-1}$ over the core region) and a slight eastward

155 shift, and the drying over southern India. The model likely underrepresents the full extent of the interaction with

156 the Western Ghats. The simulated rainfall dipole over eastern China aligns with observations, although the

157 amounts are slightly underestimated. However, the model fails to reproduce the wetting over southern China and

158 northern Indochina, possibly due to limitations in resolving interactions with the complex terrain. The sign of the

159 core precipitation anomalies over India is consistent across most of the individual ensemble members (6 out of

160 8), supporting a dominant anthropogenic origin (Supplementary Fig. S4).

161

162 Preventing European sulphate aerosols from increasing results in drier conditions over central-northern India and

163 the northern BoB, with wetting over the eastern Himalayas and Myanmar. Concurrently, the precipitation

164 anomalies over eastern Asia reverse polarity, while a pronounced precipitation deficit appears over Indochina. As

165 a result, the precipitation pattern associated with increased European $SO_2$ emissions closely resembles

166 observations and shows greater similarity than ALL across Asia. The aerosol-related precipitation response is also

167 robust across most of the ensemble members (Supplementary Fig. S4). For example, the ensemble-mean rainfall

168 trend over the core Indian region amounts to +0.33 mm day$^{-1}$ over 55 years, with increases in 6 out of 8 ensemble

169 members. This indicates that European aerosols substantially contributed to the early 20[th]-century ASM rainfall

170 trends.

171

172 The observed and simulated 55-year rainfall trends over central-northern India are also compared with the

173 corresponding range of trends from the PI experiment (Supplementary Figs. S4 and S5). The trend from CRU is

174 outside the respective 90% confidence interval, while it is slightly below in GPCC. The trend from the ALL-

175 ensemble mean is within the corresponding 90% confidence interval (+0.19 mm day$^{-1}$ over 55 years), although





five of its members exhibit positive trends that are markedly discernible from internal variability. The ensemble-
mean drying trend in the fixEU experiment exceeds the 90% confidence level. As a result, the ensemble-mean
trend associated with European aerosols (EU) is clearly distinguishable from internally-generated fluctuations,
even exceeding the 95% confidence level (+0.24 mm day$^{-1}$ over 55 years).

**3.2 Changes in the monsoon circulation**

Changes in precipitation across Asia are closely linked with variations in near-surface and atmospheric circulation
patterns (Fig. 2). The spatial correspondence between wet (dry) anomalies and regions of moisture flux
convergence (divergence) underscores the dominant role of circulation-induced moisture transport in shaping
monsoon precipitation trends (Fig. 2b). Over South Asia, a prominent anomalous anticyclone centred over
southern India and the western BoB characterises the low-tropospheric flow. Over East Asia and the western
Pacific, a zonal dipole emerges: an anomalous cyclone spans northern Indochina, southeastern China, and the
South China Sea, accompanied by an elongated anticyclone extending from central China to the western Pacific.

As Figure 2a shows, strong low-level easterlies across southern India oppose the climatological westerlies from
the Arabian Sea, leading to local rainfall deficits. Meanwhile, the climatological moisture-laden southwesterlies
are deflected northward over the northern Arabian Sea. Upon reaching central and northern India, this flow
converges with anomalous northeasterlies from northern Indochina and the northern BoB, enhancing rainfall (Fig.
2b). On the western flank of the anomalous Indo-Pakistan low, dry northerlies suppress precipitation over
Pakistan. Over East Asia, enhanced Pacific moisture transport into southern China and northeastern Indochina,
opposing the climatological southerlies linked to the western Pacific subtropical high, leads to regional wetting.
Return westerlies south of the anomalous Pacific low further reinforce the moisture flux from the BoB,
contributing to widespread wetting across the Bay and northwestern Indochina. In contrast, reduced southerly
moisture advection brings drying to southern Indonesia and northeastern China.

Figure 2c shows that the regional upper-tropospheric circulation also exhibits substantial changes, consistent with
the tropical balance among convective heating, ascent, and upper-level divergence (and vice versa). Over northern
India and the northern BoB, strong mid-tropospheric ascent coincides with rainfall enhancement, accompanied by
divergent outflow in the upper troposphere. One branch heads northeastward, converging and subsiding over
Burma and central China. Other branches are directed southwestward and southward, with corresponding mid-



tropospheric subsidence and surface anticyclones over southern India and the northern equatorial Maritime
Continent. A meridional system of divergent cells over East Asia and the western Pacific reflects the deep
convection and upper-tropospheric divergence centred near 20°N. The southern cell also converges and subsides
over the north equatorial Maritime Continent, aligning with the southern outflow from South Asia.
**3.3 Aerosol forcing over Europe**
To elucidate the generating mechanism, Figure 3a shows the widespread sulphate loading anomaly over Europe
resulting from enhanced emissions. Once emitted, aerosols are transported across and beyond the Continent.
Notably, aerosols spread southwestward towards northern Africa and the tropical Atlantic via the summertime
climatological circulation of the Azores high, and eastward over central Eurasia by the midlatitude climatological
westerlies, displacing the maximum loading and AOD (not shown) eastward of the emissions source.

Consistent with this transport, surface clear-sky downward shortwave radiation decreases across Europe, with
anomalies exceeding -3 W m$^{-2}$ over 55 years in regions of peak aerosol loading (Fig. 3b), accounting for
approximately 80% of the local reduction in the all-sky radiation (Supplementary Fig. S6). All-sky radiation
anomalies are larger (up to -5 W m$^{-2}$) due to increased cloudiness associated with circulation changes (see later).
As expected by the aerosol scattering effect, net shortwave differences between the top-of-atmosphere and surface
are minimal (not shown). Including longwave effects, the net radiative cooling at the model top reached about -2
W m$^{-2}$ over central and eastern Europe (Supplementary Fig. S6).

Cloud droplet number concentration displays widespread positive anomalies over central and eastern Europe
concurrently with a negative, albeit weak, decrease in cloud top effective radius (Supplementary Fig. S6). These
changes are consistent with those expected by the cloud response to sulphate aerosols, assuming negligible
variations in water liquid content. The latter remains relatively unchanged or slightly enhanced due to anomalous
northwesterlies from the Atlantic (see below).

As a result of the aerosol-induced dimming, near-surface temperature shows anomalous cooling over central and
eastern Europe (Fig. 3c), and concurrently, the circulation adjusts thermodynamically, with marked anticyclonic
anomalies occurring at the surface. Interestingly, Fig. 3d shows that the largest cooling and the high-pressure core
are displaced east of the aerosol loading maximum, suggesting the combined influence of direct forcing,
feedbacks, and modulation by the climatological flow. For example, eastward aerosol transport and temperature
advection by the climatological westerlies contribute in concert to displacing the aerosol cooling to the east. The
associated northeasterly flow on the eastern flank of the surface anticyclone further reinforces the cooling over
eastern Europe. Note that the simulated eastward displacement of the anomalous anticyclone is consistent with
the pattern of observed sea-level pressure trends (Supplementary Fig. S7).

**3.4 Rossby wave propagation and remote teleconnections**

As shown in Figure 4, the regional signature of European aerosols extends to the mid and upper troposphere.
Streamfunction anomalies feature a (weak) equivalent-barotropic nature and a slight northwestward tilt with
height (not shown), consistent with the mature phase of an extratropical disturbance. At 300 hPa, anticyclonic
anomalies are seen over central Europe. This pattern forms part of a wave train signal extending across the
northern hemisphere, indicating a Rossby wave response to increased European aerosols and related upper-level
relative vorticity anomalies, which serve as the primary source of wave activity.
Upper-level meridional wind anomalies align with expectations from the streamfunction/geopotential height
pattern, revealing alternating cyclonic and anticyclonic centres (Fig. 4b). The wave activity flux (e.g., Takaya and
Nakamura 2001) highlights two eastward propagation branches. The main branch extends southeastward across
the Middle East to Pakistan and northwestern India, where it weakens while turning northeastward into eastern
China and converges over Japan. This flux follows the Asian jet stream, which acts as a Rossby waveguide. A
secondary, weaker high-latitude pathway is also evident: the wave flux points northeastward towards northern
Russia, crosses northern Eurasia, and then turns southeastward over the northwestern Pacific, ultimately
converging with the main pathway.
Mid-tropospheric ascent and descent anomalies accompany the wave train (Fig. 4b), consistent with the vorticity
balance (e.g., Rodwell and Hoskins 2001). For example, the anticyclonic anomaly over Afghanistan and Pakistan
induces southward flow and subsidence to its east, suppressing precipitation over Pakistan and the western Tibetan
Plateau. Conversely, over southern China, near-surface low-pressure anomalies, upward motion, and increased
precipitation are associated with northward flow. These results suggest that key upper-tropospheric action centres,
over the Middle East and southern China, initiate low-tropospheric circulation adjustments that generate the
anomalous rainfall pattern across Asia.



### 4 Discussion and concluding remarks

The Asian summer monsoon hydroclimate underwent significant changes in the early 20th century, characterised by a wetting trend over South Asia and a southern rainfall shift over East Asia. This study finds that increased European anthropogenic sulphate aerosols played a key role in driving these observed monsoon changes. The aerosol-induced cooling and large-scale anticyclonic anomaly over central/eastern Europe extend from the surface to the upper troposphere. These anomalies trigger subsequent atmospheric circulation adjustments in the form of an eastward propagating Rossby-wave train, which is central to realising the remote aerosol impact across Asia.

These findings shed new light on the drivers and mechanisms of monsoon multidecadal variability. While most existing research has focused on recent monsoon changes, the early historical period remains unexplored. Moreover, many prior studies have relied on long, equilibrium-type experiments to enhance the signal-to-noise ratio, but these are less representative of the transient response to evolving aerosol emissions. An atmospheric propagation pathway similar to the one identified here has been discussed in earlier literature, particularly in relation to the downstream signature of the North Atlantic Oscillation (e.g., Watanabe 2004) and broader teleconnections between Europe and East Asia (e.g., Lu et al. 2002; Enomoto et al., 2003), yet it has been largely overlooked in the context of aerosol forcing (e.g., Wang et al., 2021b), which has instead typically emphasised changes in the large-scale meridional temperature gradient across Eurasia as the dominant key mechanism.

Placing our study within the broader literature on remote aerosol-monsoon interactions underscores its relevance. Cowan and Cai (2011) were among the first to identify the influence of non-Asian aerosols, primarily European sulphate, in weakening the ASM over the 20th century by inducing widespread Eurasian cooling and thereby reducing the meridional temperature gradient, which in turn weakens the southerly monsoon flow. Similarly, Guo et al. (2015, 2016) linked widespread drying across Asia to increased global aerosol emissions, mainly from non-Asian sources, and the subsequent modulation of the zonal-mean meridional temperature gradient. Focusing on the fast, atmospheric-only equilibrium response to the removal of European aerosols, Dong et al. (2016) reported a strengthening of the South Asian monsoon and a southern shift in East Asian rainfall, attributed to the downstream advection of cooler, drier air from Europe to Asia and consequent weakening of the tropospheric thermal gradient. Shawki et al. (2018) similarly found that removing European aerosol emissions strengthens, albeit weakly, the South Asian monsoon and shifts the East Asian monsoon northward, via changes in the large-



scale temperature gradient and the interhemispheric heat and moisture transport. Liu et al. (2018) reported a slight
decrease in annual mean precipitation over Asia associated with increased European sulphate aerosols, driven by
the slow (ocean-mediated) component of the total response via inter-hemispheric heating redistribution. Similar
rainfall patterns were also shown by Westervelt et al. (2018). More recently, Wang et al. (2017) and Wang et al.
(2020) highlighted that non-local aerosol emissions are as influential as local ones in weakening the East Asian
summer monsoon, primarily through easterly advection of colder air across Eurasia and resulting change in
meridional heat transport, in agreement with Dong et al. (2016).

An enhanced meridional temperature gradient between central Eurasia and the IO is a well-known contributor to
stronger monsoon precipitation over South Asia (e.g., Meehl and Arblaster, 2002). Indeed, this has been identified,
for example, as one of the elements contributing to enhanced future monsoon precipitation by the end of the 21$^{st}$
century (e.g., Meehl et al., 2024). This outcome, however, is not borne out in our study, as the mid and high
troposphere temperature averaged over the 60°-100°E sector displays enhanced cooling compared to the north-
equatorial IO, resulting in a weaker meridional temperature gradient (not shown). This discrepancy indicates that
the aerosol-induced regional monsoon circulation and precipitation variations over South Asia cannot be explained
by broad-scale sector-mean temperature changes. On the contrary, the anomalous temperature pattern over Eurasia
displays a close relationship with that induced by the anomalies in atmospheric advection, especially in the
meridional direction, in turn associated with the upper-tropospheric wave train.

This dynamical pathway and the critical role of large-scale remotely driven atmospheric dynamical changes finds
support in Bollasina et al. (2014). Although secondary to regional emissions in explaining the recent monsoon
rainfall decline, extratropical aerosols were nonetheless shown to induce widespread temperature and wind
anomalies across Asia, revealing the complex interplay between aerosol forcing, precipitation, and circulation
changes. Undorf et al. (2018a, b) also highlighted the importance of midlatitude aerosol forcing in explaining the
observed weakening of the South Asian monsoon through the mid-1970s, underscoring the key role of Eurasian-
scale dynamical adjustments.

An important question is whether Indian Ocean (IO) SSTs also influenced the monsoon. The ALL ensemble,
consistent with the overall performance of CMIP5/6 models (e.g., Roxy et al., 2014), exhibits widespread and



relatively homogeneous warming across the basin. In contrast, aerosol-induced SST trends, albeit weak, show
warming over the western equatorial IO and cooling over the subequatorial regions (Supplementary Fig. S8). This
cross-equatorial dipole in SST trends is also evident, and more pronounced, in observations (e.g., Fig. 3 in
Goswami et al., 2023), which suggests an aerosol contribution to the IO warming pattern. Further analysis of the
EU ensemble (not shown), shows that SST anomalies are largely anticorrelated with evaporation: reduced
evaporation dominates the western and equatorial IO, while increases are seen in the south. Concurrently,
anomalous near-surface divergent easterlies across the north-equatorial IO oppose the climatological (south)
westerlies, reducing evaporation and upwelling, which contribute to warming the SSTs. These patterns are similar
to those associated with the Indian summer monsoon multi-decadal variability during the 20$^{th}$ century (Goswami
et al., 2023). Also, the 55-year SST pattern resembles 20$^{th}$-century-long changes, although the latter exhibits
weaker anomalies and a less pronounced cooling (e.g., Rao et al., 2012; Roxy et al., 2014). Uncertainties remain
in identifying the causes of the persistent western IO warming due to strong feedbacks between oceanic anomalies,
atmospheric circulation, and convection (e.g., Rao et al., 2012; Swapna et al., 2014). While our findings highlight
the role of remotely-forced wind anomalies, other mechanisms may also contribute. While SST anomalies and
rainfall are anticorrelated in the long term, this relationship reverses in the early 20$^{th}$ century (Fig. 3 in Roxy et
al., 2015), aligning with our findings.

Our results have important implications. They suggest that non-local anthropogenic aerosols have contributed
significantly to shaping monsoon variability even during the early historical period. This is particularly relevant
for understanding regional-scale impacts of anthropogenic forcing, which remain uncertain due to the spatial
heterogeneity of emissions and their diverse climate effects. Importantly, under the current continued decrease in
European aerosol emissions, assuming an opposite ASM response to the one described above, remotely-driven
precipitation anomalies may significantly modulate, if not partially offset, the response to Asian aerosol emission
changes (already declining or expected to do so in the future; e.g., Lund et al., 2019; Xiang et al. 2023). For
example, assuming linearity in the combined responses, the abatement of EU sulphate aerosols would lead to a
weaker monsoon over most of South Asia, and India in particular, which is opposite to the expected wetting
associated with decreased sulphate aerosols over both South and East Asia (e.g., Bartlett et al., 2018). Conversely,
EU aerosol reductions would contribute to further amplify the reversed SFND pattern over China brought about
by decreased East Asian aerosol alone (e.g., Dong et al., 2016), resulting, for example, in enhanced drying over



southeastern China. While the future monsoon response to the projected decline in worldwide aerosol emissions
will most likely result from non-linear interactions among the different aerosol source regions, the above picture
illustrates the complex interplay among local and remote aerosols and the need for a consistent and coordinated
modelling and analysis approach (e.g., Wilcox et al., 2023).

While our analysis emphasises the central role of European aerosols, other remote emissions (e.g., from North
America) and external forcings (e.g., greenhouse gases) may have also played a role. Our conclusions are based
on ensemble experiments comprising eight members each, consistent with the minimum number typically
recommended for detecting forced multi-decadal signals (e.g., Deser et al., 2012). However, internal variability
cannot be ruled out, and studies have, for example, discussed the association of Atlantic slow-frequency variability
with the ISM multi-decadal mode (e.g., Rajesh and Goswami, 2020). In this context, the use of large single-forcing
ensembles (e.g., Smith et al., 2022; Simpson et al., 2023) offers a promising approach to disentangling the
individual contributions of greenhouse gases, anthropogenic aerosols and other external drivers. Furthermore, our
findings are based on a single climate model and therefore depend on its representation of aerosol-cloud-radiation
and circulation interactions. Given the well-known uncertainties in aerosol forcing and its climatic effects, the
response may differ across models and from real-world observations. For example, CESM1 is known to exhibit a
relatively strong aerosol effective radiative forcing (Zelinka et al., 2014; 2023), which may result in a stronger
climate response to aerosol perturbations than seen in other models.

**Acknowledgements.** MAB acknowledges support from the Natural Environment Research Council
(NE/N006038/1) and the Research Council of Norway (grant no. 324182; CATHY).

**Data availability.** GPCC data are retrieved from https://psl.noaa.gov/data/gridded/data.gpcc.html, CRU data from
https://crudata.uea.ac.uk/cru/data/hrg/, and UDEL data from
https://psl.noaa.gov/data/gridded/data.UDel_AirT_Precip.html. The ERA5 data are obtained at
https://cds.climate.copernicus.eu/datasets/reanalysis-era5-pressure-levels-monthly-means. HadISST data are
available from https://www.metoffice.gov.uk/hadobs/hadisst/, while the HadSLP2 dataset is accessed from
https://www.metoffice.gov.uk/hadobs/hadslp2/. The CESM model data used in this study are available from MAB
upon request.



**Author contribution.** WS and MAB designed the study. WS performed the analysis and completed the first draft
of the manuscript. WS and MAB discussed the results, WA, MAB, and IC edited the manuscript. GW and YL
provided suggestions on the analysis and interpretation of the results.

**Competing interests.** The authors have no competing interests to declare.



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



**Figures**

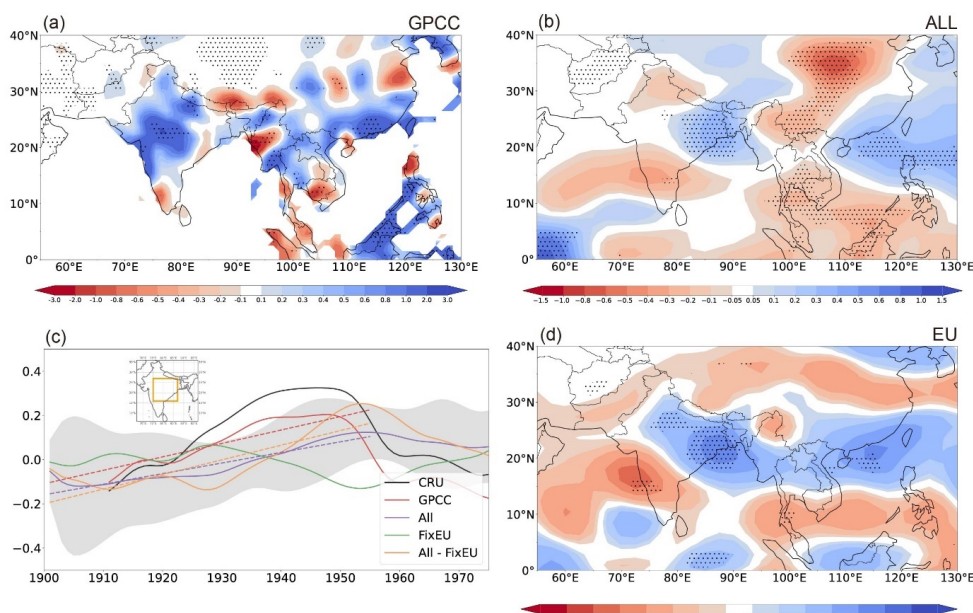



**Figure 1.** (a)-(b): Spatial patterns of the 1901-1955 linear trends of JJA precipitation (mm day$^{-1}$ (55 years)$^{-1}$) for (a) GPCC, (b) the all-forcing ensemble (ALL), and (d) the difference between ALL and the all-forcing experiment with fixed preindustrial aerosol emissions over Europe (fixEU), representing the impact of EU aerosols. The black dots mark the grid points for which the trend exceeds the 90% significance level according to the two-tailed Student's t-test. (c): Time series of area-averaged JJA precipitation anomalies (mm day$^{-1}$; deviations from the 1901-2000 climatology) over central-northern India (land-only points within 75°–87°E, 16°–27°N; area shown in inset map) smoothed with 11-year running means to highlight low-frequency (multi-decadal) fluctuations. The black and red lines represent observations (CRU and GPCC, respectively), while the purple, green and orange lines represent the ensemble means of ALL, fixEU, and their difference (EU). The grey shading represents the standard deviation of the eight-member ALL ensemble around the mean. The 1901-1955 least-squares linear trends of the simulated time series are shown as dashed lines in the corresponding colours.



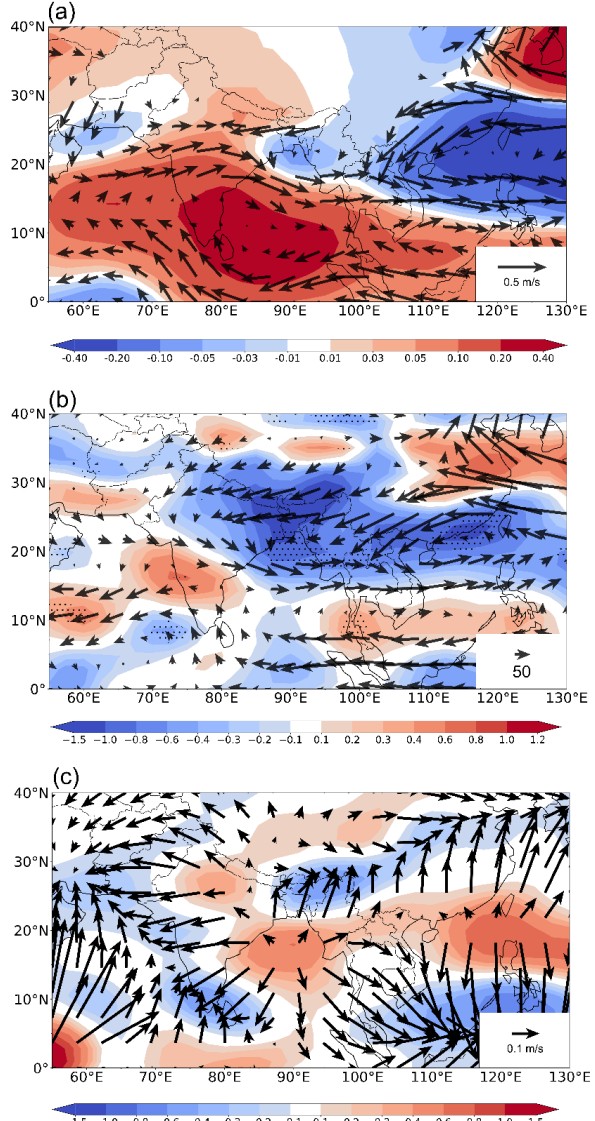


**Figure 2.** Spatial patterns of the 1901-1955 linear trends of the JJA average (a) 850-hPa winds (m s⁻¹ (55 years)⁻
¹) and 925-hPa streamfunction (colors, $10^6$ m² s⁻¹ (55 years)⁻¹), (b) 1000-300 hPa vertically integrated moisture
transport (vectors, Kg m⁻¹ s⁻¹) and its divergence (shades, mm d⁻¹ (55 years)⁻¹), (c) 150-hPa divergent circulation
(m s⁻¹ (55 years)⁻¹) and its divergence (shades; $10^{-6}$ s⁻¹ (55 years)⁻¹) associated with increased European sulphate
aerosols (difference between the ALL and fixEU ensemble means). The black dots in (b) mark the grid points for
which the trend exceeds the 90% significance level according to the two-tailed Student's t-test.




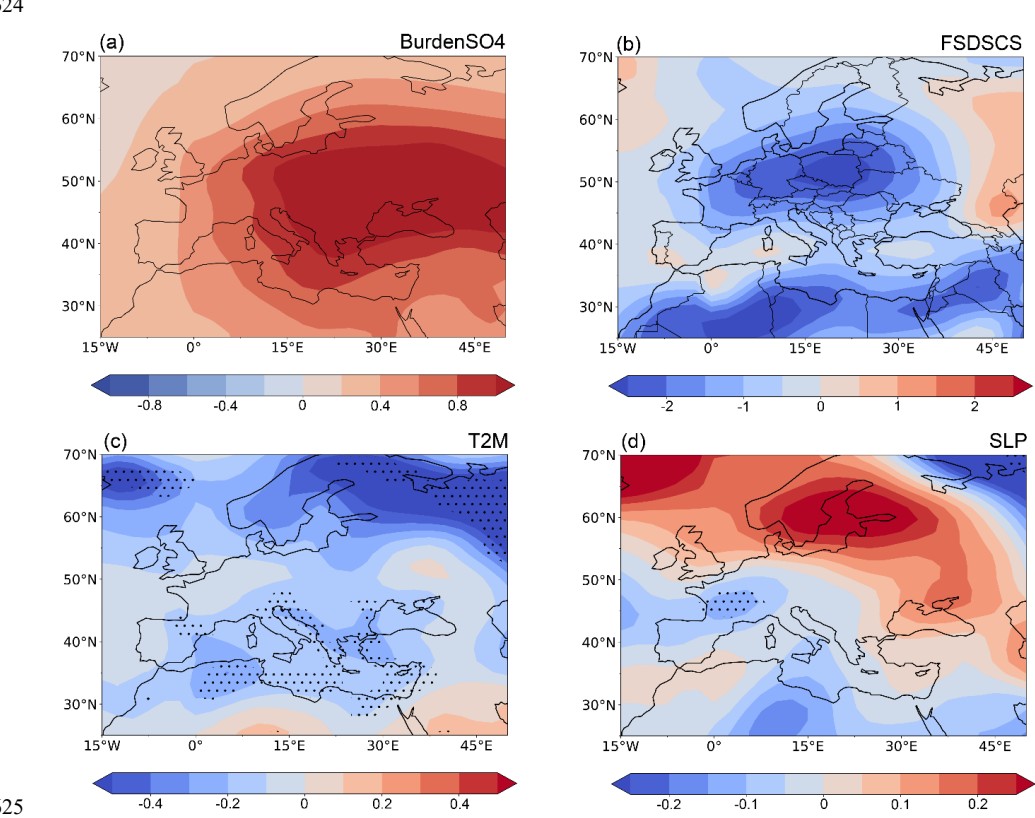



**Figure 3.** Spatial patterns of the 1901-1955 linear trends of the JJA average (a) column sulphate aerosol burden
($10^{-5}$ kg m$^{-2}$ (55 years)$^{-1}$), (b) surface clear-sky downward shortwave radiation (W m$^{-2}$ (55 years)$^{-1}$), (c) 2-m air
temperature (K (55 years)$^{-1}$), and (d) sea level pressure (hPa (55 years)$^{-1}$) associated with increased European
sulphate aerosols (difference between the ALL and fixEU ensemble means). The black dots in (c) and (d) mark
the grid points for which the trend exceeds the 90% significance level according to the two-tailed Student's t-test.




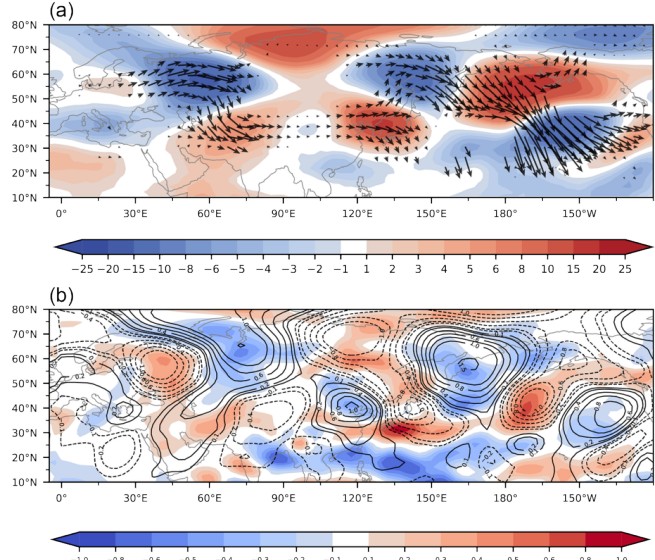


**Figure 4.** Spatial patterns of the 1901-1955 linear trends of the JJA average (a) 300-hPa wave activity flux
(vectors; $10^{-4}$ m² s⁻² (55 years)⁻¹) and streamfunction (shades; $10^5$ m² s⁻¹ (55 years)⁻¹), and (b) 300-hPa meridional
wind (contours; m s⁻¹ (55 years)⁻¹) and 500-hPa vertical velocity (shades; $10^{-2}$ Pa s⁻¹ (55 years)⁻¹) associated with
increased European sulphate aerosols (difference between the ALL and fixEU ensemble means).