# Peer review of "European sulphate aerosols were a key driver of the early"

_EGUsphere, 2025_

## Author Comment (AC1)

We thank both reviewers for their helpful comments and constructive suggestions. Below we provide our point-by-point responses below (in bold).

**Referee 1**

**(1) The authors state in lines 152-153 that "the ALL ensemble mean captures the main spatio-temporal characteristics of the observed precipitation trends". I interpret this to mean that Figure 1a and 1b share some similarity. Visually I am not convinced that I see this. Firstly, the two figures are not plotted on the same color bar, so all of the changes in the ALL ensemble are much weaker than those from GPCC. Additionally, it does not appear to me that the locations with the strongest positive and negative anomalies are collocated. Is there some spatial correlation between the maps in Figures a and b? That would help to solidify this argument.**

Thank you for raising the point. Our comparison focuses on the spatial pattern of the anthropogenically-forced response in South Asian summer precipitation. Observed changes reflect both external forcing (primarily, but not entirely, anthropogenic) and internal climate variability, and therefore exhibit a wider range of magnitudes and more pronounced spatial variability than an ensemble of coarser-resolution climate model simulations. On the other hand, station-based observational datasets are also not entirely accurate, particularly in the early 20th century, because of sparse and non-uniform measurement networks and data-quality limitations—issues that are especially pronounced over complex terrain and at small spatial scales. This can result in overly inhomogeneous trend patterns, influenced by a non-uniform data distribution. As a result, we expect a free running model to show biases in the response compared to observations, especially when focusing on sub-regional rainfall features. Because the ensemble-mean trend in CESM is generally weaker than in observations, using a common observation-based colour bar would compress the model values and obscure their spatial structure. To preserve clarity and ensure the spatial patterns remain interpretable, we therefore adjusted colour scales for model results. However, in response to a similar comment by Reviewer 2, we now explicitly highlight in both the main text and the caption of Figure 1 that different contour intervals are used (with the interval for the simulated changes being halved relative to that used for the observations).

In our case, for example, the CESM ALL ensemble mean indeed shows an eastward shift of the centre of positive precipitation anomalies over the Indian subcontinent relative to observations. However, the model pattern shows the key large-scale features of the observed changes, namely enhanced summer rainfall over central/northern India and South China, and drying over southern India and the southern Indochina Peninsula. Note that the EU-only forced pattern is closer to observations than ALL, with in particular the zonally-oriented wetting across northern India, northern Indochina and

southern China well captured by the ensemble. This suggests a partial offset of the aerosol-driven changes by GHGs in the combined (ALL) response, possibly due to the relatively high TCR and ECS in CESM1 (2.3K and 4.1K, respectively) compared to other CMIP5 models (Andrews et al., 2012; Meehl et al., 2013).

Tu put this into a broader context, we have examined the spatial distribution of the linear precipitation trend in a set of available CMIP6 climate models. As Figure R1 shows, there is a wide inter-model spread even the simulation of the large-scale features. For example, many of the models fail to capture the wetting over northern India and southeastern China, with some models even simulating drying.

While we acknowledge some limitations in the CESM1 simulated rainfall pattern, we believe spatial correlation to be overly sensitive to the precise location of the features being simulated, which is particular critical when we compare 55-year trends. The model might predict a trend pattern that is qualitatively correct but slightly displaced, which would result in a low or misleading spatial correlation value even if the model's overall spatial structure is reasonable. This is even more accentuated for variables such as precipitation which has inherently small spatial correlation scales.

Andrews, T., et al., 2012: Forcing, feedbacks and climate sensitivity in CMIP5 coupled atmosphere–ocean climate models. Geophys. Res. Lett., 39, L09712, doi:10.1029/2012GL051607.
Meehl, G. A., et al., 2013: Climate Change Projections in CESM1(CAM5) Compared to CCSM4. J. Climate, 26, 6287–6308, https://doi.org/10.1175/JCLI-D-12-00572.1.

[Figure]

**Figure R1.** Spatial patterns of the 1901-1955 linear trend of JJA precipitation (mm day$^{-1}$ (55 years)$^{-1}$) in the historical (all forcing) experiments for a set of CMIP6 climate models (identified by the title above each panel) and the multi-model ensemble mean (MME). Each of the model trend is the ensemble mean of at least 3 ensemble members. The colour interval matches the one used for Figure 1 in the main text.

**There are numerous times throughout this manuscript that references "not shown" results (for example, line 214, 221, 325). Any results that are important enough to**

**mention in the manuscript should be at least shown in a supplement.**

We thank the reviewer for this suggestion. We have added several variables to the Figures in the Supplementary Material. Figure S6 also includes (g) Aerosol Optical Depth (AOD), and (h) net radiation at the surface. Also, Figure S8 now displays (a) SST (K) and (b) Evaporation (mm day$^{-1}$). The revised Figures S6 and S8 are copied below.

[Figure]

**Figure S6**. Spatial patterns of the 1901-1955 linear trends of JJA (a) cloud droplet effective radius ($10^2$ μm (55 years)$^{-1}$), (b) vertically-integrated cloud droplet concentration (m$^{-2}$ (55 years)$^{-1}$), (c) net radiation at the model top (W m$^{-2}$ (55 years)$^{-1}$, (d) total column liquid water path ($10^3$ Kg m$^{-2}$ (55 years)$^{-1}$), (e) surface all-sky downward shortwave radiation (W m$^{-2}$ (55 years)$^{-1}$), (f) net radiation at the surface (Wm$^{-2}$ (55 years)$^{-1}$), (g) Aerosol Optical Depth (AOD) and (h) net radiation at the top of atmosphere (W m$^{-2}$ (55 years)$^{-1}$) associated with increased European sulphate aerosols (difference between the ALL and fixEU ensemble means).

[Figure]

**Figure S8.** Spatial patterns of the 1901–1955 changes of (a) JJA sea-surface temperature (K) and (b) evaporation (mm day$^{-1}$) difference between ALL and the all-forcing experiment with fixed preindustrial aerosol emissions over Europe (fixEU), representing the impact of EU aerosols. Changes are calculated as the difference between the (1941–1955) and the (1901–1915) averages.

**There are also a couple times where later figures are referenced by the statement "see below" (for example line 219, 228). It would benefit the reader to instead reference a specific figure or section.**

**The figure referencing is at times inconsistent. Sometimes it is Figure X, and others Fig. X. One should be used consistently throughout the manuscript.**

Thank you, manuscript revised.

**Referee 2**

**Line 60. "NH", please spell it out when it appears first time in the paper.**
Thank you, manuscript revised.

**Line 77. "driving monsoon" to "driving Asian summer monsoon".**
Thank you, manuscript revised.

**Lines 162-170. It is better to add a comment to compare trends between observations and model simulations.**
Thank you. We have more clearly highlighted the comparison in the paragraph.

**Line 163. "BoB" has not defined yet.**
Thank you, manuscript revised.

**Lines 172-179. What are grey bars in two observations in Supplementary Fig.S4? From text, it seems that they are trends based on PI control. If they are, why are the two grey bars in GPCC and CRU are not the same as those shown in ALL and fixEU?**
Yes, in all cases the grey bars represent the trends derived on the PI experiment. However, the PI simulation is sampled differently for observations and for the modelled trends, because observations represent a single realisation of the climate evolution, whereas the simulations comprise an ensemble of eight realisations. For the observational comparison, the probability distribution of unforced linear trends is computed from 146 (200-55+1) overlapping 55-year segments. For the model ensemble, those same 146 samples are additionally grouped into random averages of 8 (matching the ensemble size) selected from a distribution of $\sim 10^{10}$ samples.

**Lines 172-179. There is no description about Supplementary Fig. S5.**
Thank you, manuscript revised.

**Figure 1 caption. It is better to emphasize that the color scale of precipitation trends in GPCC and model simulations are different.**
Thank you. In response to a similar comment by Reviewer 1, we now explicitly highlight in both the main text and the caption of Figure 1 that different contour intervals are used (with the interval for the simulated changes being halved relative to that used for the observations).

**Figure 2b. Please use same color scale for vertically integrated moisture transport divergence trend and precipitation trend shown in Figure 1d for easy comparison.**
Figure 2b has been revised accordingly. The revised Figure 2 is copied below.

[Figure]

**Figure 2.** Spatial patterns of the 1901-1955 linear trends of JJA average (a) 850-hPa winds (m s$^{-1}$ (55 years)$^{-1}$) and 925-hPa streamfunction (colors, $10^6$ m² s$^{-1}$ (55 years)$^{-1}$), (b) 1000-300 hPa vertically integrated moisture transport (vectors, Kg m$^{-1}$ s$^{-1}$) and its divergence (shades, mm d$^{-1}$ (55 years)$^{-1}$), (c) 150-hPa divergent circulation (m s$^{-1}$ (55 years)$^{-1}$) and its divergence (shades; $10^{-6}$ s$^{-1}$ (55 years)$^{-1}$) associated with increased European sulphate aerosols (difference between the ALL and fixEU ensemble means). The black dots in (b) mark the grid points for which the trend exceeds the 90% significance level according to the two-tailed Student's t-test.

**Line 214. "AOD". Please spell it out when it appears first time.**
Thank you, manuscript revised.

**Lines 216-218. Please use the same color scale for Fig. 3b and Supplementary Fig.**

**S6e to these descriptions.**

Figure S6e has been revised accordingly. Additionally, to address a comment from Reviewer 1, we added to the original Figure S6 two variables (only mentioned, but not shown, in the previous submission): (g) Aerosol Optical Depth (AOD), and (h) net radiation at TOA. The revised Figure S6 is copied below.

[Figure]

**Figure S6**. Spatial patterns of the 1901-1955 linear trends of JJA (a) cloud droplet effective radius ($10^2$ μm (55 years)$^{-1}$), (b) vertically-integrated cloud droplet concentration (m$^{-2}$ (55 years)$^{-1}$), (c) net radiation at the model top (W m$^{-2}$ (55 years)$^{-1}$, (d) total column liquid water path ($10^3$ Kg m$^{-2}$ (55 years)$^{-1}$), (e) surface all-sky downward shortwave radiation (W m$^{-2}$ (55 years)$^{-1}$), (f) net radiation at the surface (Wm$^{-2}$ (55 years)$^{-1}$ ), (g) Aerosol Optical Depth (AOD) and (h) net radiation at the top of atmosphere (W m$^{-2}$ (55 years)$^{-1}$) associated with increased European sulphate aerosols (difference between the ALL and fixEU ensemble means).

**Lines 237-238. The magnitude of SLP trend is only about 10% of those based on observations or CESM ALL simulation. This implies that SLP trends over Europe in CESM ALL might not be resulted from European sulphate aerosol emissions. Please comment.**

While our study focuses on the forced response to aerosol changes, we agree with the reviewer that other forcing factors (e.g., GHGs) and internal climate variability can also contribute. Regarding the SLP trend pattern, the simulated magnitude of the anticyclone in the ALL ensemble is comparable to that in the observations. The main discrepancy lies in its northward displacement in ALL. To provide a more complete picture, we now also show the SLP trend from the EU ensemble in the revised Figure S7 in the Supplementary Material. Although this ensemble still exhibits a northward shift of the positive SLP anomalies, the core of the anticyclone is more clearly defined and positioned much closer to the observed location. As illustrated in the Figure R2 below,

most SLP features are consistently reproduced across the majority of individual ensemble members, indicating that internal variability plays a role but does not dominate the simulated response. However, it is important to note that observations themselves are not very reliable in the early 20th century. For example, Figure R3 shows the average number of observations per month within each 5°x5° grid cell used to construct the HadSLP dataset for the entire 1901-1910 period (data from https://www.metoffice.gov.uk/hadobs/hadslp2/data/download.html). While central-western Europe is relatively well-sampled, the scarcity of data across central-eastern and northern Europe is striking, leading to substantial uncertainty in the observed trend pattern. In light of this, comparison between simulated and observed trends should focus on the large-scale SLP features and their consistency with the expected physically-based understanding —for instance, the broad westward shift of the core anticyclonic anomalies—rather than on finer-sub-regional scale details.

[Figure]

**Figure R2.** Spatial patterns of the 1901-1955 linear trend of JJA sea-level pressure (hPa (55 years)$^{-1}$) in the all-forcing ensemble (ALL) and the differences between ALL and the all-forcing experiment with fixed pre-industrial aerosol emissions over Europe, representing the impact of EU aerosols. Hatching mark grid points for which 5 out of 8 members agree on the sing of the trend. The colour interval matches the one used in Figure S7.

[Figure]

**Figure R3.** Spatial patterns of the average number of observations per month within each 5°x5° grid cell used to construct the HadSLP dataset over the 1901-1910 period.

**Lines 256-262. The wave train shown in Fig.4 is in mid-high latitude while anomalous ascents and precipitation are over 10-20N across India into southern China and northeastern Indochina. It is therefore not clear how does the wave train trigger these anomalous ascents and precipitation. Fig.2 shows circulation changes over the western tropical Pacific induced by European sulphate aerosol. The reviewer wonder whether there is any local feedback between ocean and atmosphere.**

The reviewer is correct. We briefly mentioned the interaction between the wave-induced upper-tropospheric anomalies over southeastern China and the western Pacific and the associated anomalous rainfall anomalies at the end of section 3.4. The wave propagates eastward across Asia along ~30°-40°N. In the upper troposphere, an anomalous anticyclone develops over eastern China and the western Pacific (120°-140°E). Northward flow and ascent occur on its western flank, while subsidence is found to the east, in agreement with the fundamental vorticity balance. At lower levels, this configuration generates an anomalous westerly flow from the western Pacific toward south-eastern China, enhancing moisture transport and producing a positive precipitation anomaly. This rainfall anomaly, in turn, drives strong variations in the tropical divergent circulation: part of the divergent outflow extends toward the Maritime Continent, where it converges, descends, and produces anomalous anticyclonic flow. The interaction between extratropical wave-induced anomalies over eastern China/western Pacific and the resulting meridional adjustments in the tropical divergent circulation account for the characteristic tripolar anomalous rainfall pattern across East Asia. We have slightly rephrased the paragraph in the manuscript to improve the clarity.